

# Observation and explanation of spurious seismic signals emerging in teleseismic noise correlations

Lei Li[1,2], Pierre Boué[2], Michel Campillo[2]

[1]State Key Laboratory of Earthquake Dynamics, Institute of Geology, CEA, Beijing, China
[2]Université Grenoble Alpes, CNRS, IRD, IFSTTAR, ISTerre, Grenoble, France

*Correspondence to*: Lei Li (lilei@ies.ac.cn)

**Abstract.** Deep body waves have been reconstructed from seismic noise correlations in recent studies. Authors prospect their great potentials in deep-Earth imaging. In addition to the expected physical seismic phases, some spurious arrivals having no correspondence in earthquake seismograms are observed from the noise correlations. The origins of the noise-

derived body waves have not been well understood. Traditionally, the reconstruction of seismic phases from inter-receiver noise correlations is attributed to the interference between waves from noise sources in the stationary-phase regions. The interfering waves emanating from a stationary-phase location have a common ray path from the source to the first receiver. The correlation operator cancels the common path and extracts a signal corresponding to the inter-receiver ray path. In this study, with seismic noise records from two networks at teleseismic distance, we show that noise-derived spurious seismic

signals without correspondence in real seismograms can arise from the interference between waves without common ray path or common slowness. These noise-derived signals cannot be explained by the traditional stationary-phase arguments. Numerical experiments reproduce the observed spurious signals. These signals still emerge for uniformly distributed noise sources, and thus are not caused by localized sources. We interpret the presence of the spurious signals with a less restrictive condition of quasi-stationary phase: providing the time delays between interfering waves from spatially distributed noise

sources are close enough, the stack of correlation functions over the distributed sources can still be constructive as an effect of finite frequencies, and thereby noise-derived signals emerge from the source averaging.

## 1 Introduction

The technique of noise correlation is implemented simply via computation of correlation functions between ambient noise records at receivers. Theoretical and experimental studies (e.g., Lobkis and Weaver, 2001; Wapenaar, 2004) have shown that

under restrictive conditions, the inter-receiver correlation function converges toward the response that would be recorded at one receiver if a source was located at the other. This is, by definition, the Green's function of the medium between the two receivers. Great achievements have been realized with the introduction of the noise correlation technique into solid-Earth seismology (Campillo and Paul, 2003; Shapiro and Campillo, 2004). The most common applications are passive imaging (e.g., Sabra et al., 2005; Shapiro et al., 2005) and monitoring (e.g., Brenguier et al., 2008; Wegler et al., 2009) of subsurface



using signals derived from seismic noise. We refer to (Campillo and Roux, 2015) for a systematic review on the recent progress in the theoretical and methodological aspects, and the various noise-based applications.

Both the surface-wave and body-wave parts of the Green's function can be reconstructed from noise correlations. Surface waves are easier to extract due to their dominance in the noise power spectra. There are relatively few, yet promising, examples of noise-derived body waves. Recently, it has been demonstrated that deep body-wave signals that propagate through the mantle and core can be extracted from ambient noise correlations (e.g., Boué et al., 2013b, 2014; Nishida, 2013). In contrast to previous studies that primarily discussed the reconstruction of normal seismic phases from noise correlations, we focus our analysis here on the interpretation of a spurious seismic phase that can be observed from noise correlations between receivers separated at teleseismic distances. A seismic phase is termed normal if it is present in the Green's function of the medium, and spurious if it is not observed in real seismograms and does not obey the theory of seismic wave propagation. This paper is organized as follows. In Sect. 2, we describe the processing of seismic noise data. In Sect. 3, we correlate the processed noise records and show the observation of noise-derived spurious arrivals emerging earlier than the direct P waves. In Sect. 4, we develop a new double-array technique to estimate the slownesses of the interfering waves and analyse the origin of the observed spurious phase. In Sect. 5, we propose a mechanism to explain the generation of the spurious phase. More elaborate numerical experiments are implemented in Sect. 6 to reproduce the observed signals and to investigate the sensitivity to the distribution of seismic noise sources.

## 2 Noise data processing

Continuous seismograms recorded in 2008 by the broadband stations of the Full Range Seismograph Network of Japan (the FNET array) and the northern Fennoscandia POLENET/LAPNET seismic network in Finland (the LAPNET array), are used in this study to calculate the double-array noise correlations (Fig. 1). The aperture of the LAPNET array is ~700 km, and that of the FNET array is nearly 1,400 km. There are 1,558 FNET-LAPNET station pairs in all. The distance between the FNET and LAPNET stations ranges from ~56° to 70°, with a centre-to-centre distance of 63°.

The processing of seismic noise data is segment-based, as demonstrated in Fig. 2. The processing is similar to that adopted by Poli et al. (2012) and Boué et al. (2013b). First, routine signal-processing operations are applied to the raw seismograms (including mean and linear trend removal, bandpass filtering, 5 Hz down-sampling, instrumental response deconvolution). Then, continuous seismograms are divided into 4-h segments. The frequency spectra of the segments are whitened between periods of 1 s and 100 s. The spectral whitening removes amplitude information and retains only the phase spectrum. One may further clip the spectral-whitened waveform at several times of the standard deviation to reduce large transients. A selection filter is applied to the segments to detect and reject those containing transient impulses like earthquakes and electronic glitches.





In the previously mentioned studies, the selection filter was based on the energy ratio between segment and daily trace, which can be deemed as a coarse version of the classic STA/LTA method commonly used for earthquake detection (Allen, 1982). Here, we adopt a new kurtosis-based selection filter. The kurtosis, or fourth-order normalized central moment, is defined as $\kappa = \mathbf{E}[s^4]/(\mathbf{E}[s^2])^2 - 3$, with $\mathbf{E}[\cdot]$ the expectation operator and $s$ the demeaned waveform. It is highly sensitive to impulsiveness (Westfall, 2014), close to zero for stationary noise but increasing abruptly in the presence of transient impulses (see Fig. 3 for examples). The selection filter can be applied at any stages of the processing (raw, whitened, clipped waveforms). In this study, segments of kurtosis beyond 1.5 are discarded. The kurtosis has also been used in detecting earthquakes and picking phases (e.g., Baillard et al., 2014; Saragiotis et al., 2002) and is first used for noise data processing. Compared to the energy-based selection, the kurtosis-based selection depends on the statistics of the segment itself and is more robust when the strength of seismic noise changes rapidly.

## 3 Observation of spurious phase

We correlate all of the available pairs of processed noise segments and stack them to produce the correlation function of the year-long data for each FNET-LAPNET station pair. The computation of correlation function is diagrammatized in Fig. S1. The correlation function contains a causal part and an acausal part that correspond to the positive and negative time lags, respectively. In this paper, the acausal correlations correspond to seismic waves that travel from FNET to LAPNET (causal: from LAPNET to FNET).

The noise correlations of all of the FNET-LAPNET station pairs are binned in an inter-station distance interval of 0.1°, to produce the waveform sections for the acausal and causal parts of the noise correlations. The filtered sections (periods of 5 s to 10 s) of the vertical-vertical noise correlations are shown in Fig. 4 and the broadband sections in Fig. S2. The theoretical P and PcP waves marked on the panels are predicted using the Taup program (Crotwell et al., 1999) and the IASP91 Earth model (Kennett and Engdahl, 1991). A coherent arrival between 410 s and 450 s is clearly visible in the acausal section of noise correlations, about 200 s earlier than the direct P wave that should be the primary arrival. The early arrival has no correspondence in the true Green's function of the Earth medium, and thereby is undoubtedly a spurious phase. Spectral analysis indicates that the spurious phase has a peak period of 6.2 s (Fig. S3), typical for secondary microseisms. As estimated from the acausal vespagram in Fig. 4, the emerging time and apparent slowness of the spurious phase at 63° distance are about 430 s and 4.6 s/deg, respectively. The spurious phase is only observed in the vertical-vertical noise correlations, indicating that it is likely a P-type phase. In the causal correlations, a corresponding spurious phase is hardly discriminable.



## 4 Origin of signals from P-PKPab correlations

In the previous section, a prominent spurious phase was observed in the FNET-LAPNET noise correlations, and its apparent slowness and emerging time were estimated. The double-array configuration offers the possibility to estimate the respective azimuths and magnitudes of the slownesses of the correlated wavefields that should be responsible for the spurious phase.

Given a wave with slowness $\boldsymbol{p}^A$ at FNET and a wave with slowness $\boldsymbol{p}^B$ at LAPNET, the time difference between the time delay between the $i$th FNET station and the $j$th LAPNET station and the center-to-center reference time can be determined from Eq. (1):

$$\Delta t_{ij} = \boldsymbol{x}_i^A \cdot \boldsymbol{p}^A - \boldsymbol{x}_j^B \cdot \boldsymbol{p}^B \tag{1}$$

where $\boldsymbol{x}$ are the local coordinates of the station with respect to the array center, and superscripts $A$ and $B$ distinguish between

FNET and LAPNET. For a given pair of $(\boldsymbol{p}^A, \boldsymbol{p}^B)$, the noise correlations of all station pairs are beamed by Eq. (2):

$$B(t, \boldsymbol{p}^A, \boldsymbol{p}^B) = \langle\, C_{ij}\big(t + \Delta t_{ij}\big) \rangle \tag{2}$$

Where $\langle \cdot \rangle$ denotes ensemble average and $C_{ij}$ is the correlation function between the $i$th FNET station and the $j$th LAPNET station. This delay-and-sum process for the double-array data is known as the double-beam method, which has been applied to earthquake data (e.g., Krüger et al., 1993; Rost and Thomas, 2002) and noise correlations (e.g., Boué et al., 2013a; Roux

et al., 2008). Repeating the double-beamforming for a range of $\boldsymbol{p}^A$ and $\boldsymbol{p}^B$, the power map of the double-beamed waveforms $\langle |B(\boldsymbol{p}^A, \boldsymbol{p}^B)|^2 \rangle$ provides the optimal slowness estimates for the interfering waves. Here we call this method the double-array slowness analysis.

To investigate the resolution capability of the double-array slowness analysis for the FNET-LAPNET geometry, we make

numerical experiments by presuming (a) the same slowness at FNET and LAPNET (4.6 s/deg), and (b) different slownesses at FNET (4.7 s/deg) and LAPNET (4.2 s/deg). The results are plotted in Fig. 5. In both cases, the slownesses of the correlated waves at FNET and LAPNET are well resolved. The results for the observed spurious phase are shown in Fig. 6a. The azimuths of the correlated waves responsible for the spurious phase are confined to the great-circle direction across FNET and LAPNET, implying that the corresponding microseism noise source should be located on the great circle. The

slowness at FNET is distinct from that at LAPNET. The 4.7 s/deg slowness at FNET is valid for deep mantle phases, while the 4.2 s/deg slowness at LAPNET is characteristic of core phases. Numerical experiments in Fig. 5 have justified the reliability of this slowness discrepancy estimation. To investigate if this discrepancy is caused by the lateral heterogeneity of structure beneath the seismic networks, we apply the double-array slowness analysis to the acausal P waves in the FNET-LAPNET correlations as reference (Fig. S4). If lateral heterogeneity causes the slowness discrepancy for the spurious phase,

one should also observe a similar phenomenon for the inter-receiver P wave. It is revealed that the slownesses of the interfering waves for the P wave coincide with each other and are close to the predicted value (6.7 s/deg in IASP91 model). The P-wave results again justify the reliability of our slowness estimates, and indicate that lateral heterogeneity is not the reason for the slowness discrepancy observed from Fig. 6a.





The peak period of the spurious phase is a typical value for secondary microseisms, which are dominantly excited by ocean wave-wave interactions (Hasselmann, 1963; Longuet-Higgins, 1950). It implies that the noise sources are mainly distributed on the global ocean surface. We propose a slowness-track method to identify the ray paths of the interfering waves from source to receivers that produce the spurious phase (Fig. 6b). All the body phases that are discernible in the vertical-component earthquake seismograms are considered as candidates (see labels in Fig. 7). For a specific seismic phase, the distance from source to receiver can be derived from the slowness. The pairs of seismic phases are rejected if the difference between the distances from the source to the receivers differs from 63° or if their time delay deviates from 430 s. For clarity, only several typical P-type phases (P, PcP, PP, and PKP branches) are shown in Fig. 6b. Finally, we find that the correlation between the P wave at ~89° distance and the PKPab wave at ~152° distance is the only combination that satisfies all the constraints. As can be seen from Fig. 6b, at 89° distance, the PcP wave arrives almost simultaneously with the P wave, suggesting that PcP-PKPab also has a time delay of ~430 s at 63° inter-receiver distance. There are also other pairs of seismic phases meeting the constrains other than the slowness estimates. The slownesses estimates are crucial for the exclusive determination of the interfering waves.

It is logical that the spurious phase is observable in the vertical-vertical noise correlations only, as the correlated waves are both P-type and their amplitudes are dominantly projected onto the vertical components (the lower quality of the horizontal components could be another reason). From Fig. 6, one can locate the source responsible for the acausal and causal spurious phase, at around [45°S, 174°E] and [12°S, 28°W], respectively. Recall that the spurious phase is not observable in the causal correlations. Comparisons with global oceanic microseism noise sources (Fig. 8) indicate that this time asymmetry can be explained by the difference in the strength of the acausal and causal noise sources: the acausal source in the ocean south of New Zealand is energetic, whereas the causal source in the low-latitude Atlantic east of Brazil is much weaker.

## 5 Explanation of quasi-stationary phase

The observed spurious phase originates from the correlation between teleseismic P waves and PKPab waves that emanate from the oceanic microseism noise source south of New Zealand. In this section, we explain how such spurious signals can be generated from the interference between waves having distinct slownesses and no common path. Considering ambient noise wavefield as a superposition of waves from uncorrelated sources distributed on Earth's surface (Fig. 9a), the correlation function between the noise records at two receivers is equivalent to a stack of the correlation functions for individual sources (i.e., source averaging). We first simulate the source-wise correlation functions by convolving a wavelet of 6.2 s period with the time delays between the two correlated seismic phases. The final inter-receiver correlation function is obtained from a stack over all sources. In this ray-based simulation, amplitude information is neglected for simplicity. The





result for the source averaging of the P-PKPab correlations is shown in Fig. 9b, while that of the P-PKPbc correlations is shown in Fig. 9c for comparison.

The construction of seismic signals from noise correlations has been usually explained with the stationary-phase condition
(e.g., Wapenaar et al., 2010). We show such an example in Fig. S4, which is the reconstruction of the inter-receiver P wave from the correlation between the P and PP waves. The P-wave reconstruction is linked to the presence of the extreme (stationary) point on the curve of the P-PP time delay. The P and PP waves from the source at the stationary-phase location (Fig. S4, source A) have a common path and a common slowness. However, as for the spurious phase observed between FNET and LAPNET, the correlated P and PKPab waves have no slowness or ray path in common, and there is no stationary
point on the curve of the P-PKPab time delay (Fig. 9b). The strict stationary-phase condition is not fulfilled, and thus the emergence of the spurious phase cannot be explained by this argument. Despite the missing of stationary points on the curve of the P-PKPab time delay, the interference between finite-frequency P and PKPab waves is constructive over the shaded range in Fig. 9b. That leads to the presence of a pulsive signal in the final inter-receiver correlation by source averaging. In contrast to the strict condition of stationary phase, we propose to call this mechanism the condition of quasi-stationary phase,
and refer to this range of sources as the quasi-stationary-phase region or effective source region. At short periods (1 s or shorter), numerical tests indicate that source averaging for the P–PKPab correlations can still produce signals, with narrower effective source region shrinking toward larger source-receiver distances. The concentration of the spectral content of the spurious phase in the band of secondary microseisms is resultant from the distribution of spectral content of seismic noise sources. Secondary microseisms correspond to the largest peak in the seismic noise spectra (Peterson, 1993).

Experiences from earthquake observations indicate that PKPbc waves are generally the dominant PKP branch at distances from ~144° (the PKP-wave caustic) to ~153° (Kulhánek, 2002). Microseism studies have also reported that PKPbc waves can be more prominent (e.g., Gerstoft et al., 2008; Landés et al., 2010). However, our analysis reveals that the spurious phase originates from the interference of P waves with PKPab waves, rather than with PKPbc waves. From the source-averaging
experiment for the P-PKPbc correlations (Fig. 9c), one can see that the P-PKPbc time delay varies almost linearly against the source-receiver distance, and that the dynamic range of the time delay is broad. Consequently, the signals in the source-wise correlations are out of phase, which leads to a destructive source averaging.

## 6 Effect of source distribution

The ray-based simulation above is simple to implement and efficient to reveal the origin of the spurious phase. It uses only
the phase information and neglects the effect of amplitude variations. This simplification ensures that the spurious phase is not likely to be caused by a strong localized source. To confirm, in this section, we implement a formal wave-based simulation to show that the observed spurious arrivals can be well reproduced under ideally uniform source distribution. As





shown in Fig. 8, the spatial variations of the power of global microseism sources are heavily fluctuated. The spurious phase is observable in the acausal correlations because the corresponding source is strong, and is hardly observable in the causal correlations because the responsible source is too weak. It is worth confirming whether the spurious phase can be eliminated with an ideally uniform source distribution or not.

We request the vertical components of the synthetic global broadband seismogram for the *iasp91_2s* model from the IRIS Syngine Data Service (Krischer et al., 2017) powered by the spectral-element program AxiSEM (Nissen-Meyer et al., 2014) and the Python packages Obspy (Krischer et al., 2015) and Instaseis (Van Driel et al., 2015). A mask is applied to the full waveforms (Fig. 10a) to extract the P waves and the PKPab waves (Fig. 10b). Providing that the uncorrelated noise sources

are distributed evenly on the global surface, we compute the source-wise correlations and stack them for each inter-station distance, using the data in Fig. 10b. A global section of synthetic P-PKPab correlations is obtained accordingly (Fig. 10c). The spurious phase is clearly reproduced, which suggests that it is not caused by unevenly distributed noise sources.

Repeating the ray-based simulation in Fig. 9b for various inter-receiver distances, one can also obtain a full section of the P-

PKPab correlations. Due to the neglect of amplitude information, the ray-based simulation over-estimates the observable range of inter-receiver distances for the spurious phase. The wave-based simulation in this section is undoubtedly more realistic. The theoretical time-distance curve of the spurious phase can be picked from the synthetic sections. It is almost identical for the ray- and wave-based simulations in the observable distance range, and fits well with the observed spurious arrivals in Fig. 4. We also compare the simulations for two-dimensional plane model (sources along the great circle) and

three-dimensional spherical model (sources on global surface). The picked time-distance curves are the same, while the amplitudes of signals are different. In the case of uniformly distributed sources, it is safe to implement two-dimensional simulation for efficiency.

## 7 Discussions and conclusions

We observe early spurious arrivals in the teleseismic noise correlations between the Japan and Finland stations. These

signals are prominent and isolated from other strong seismic phases, making it a good agency to unveil the generation mechanism of such spurious phases. We provide evidence in support of that the observed signals origin from the interference between ballistic P waves and PKPab waves that emanate from oceanic microseism noise sources south of New Zealand. The interfering waves have no ray path or slowness in common, and do not meet the traditional condition of stationary phase. The effective source location responsible for the spurious phase does not correspond to a stationary point. We propose a less

restrictive condition of quasi-stationary phase, which explains our finite-frequency observations.



The interfering P and PKPab waves have deterministic ray paths that sample the deep mantle and the core. We expect a way to use the spurious phase to investigate the deep Earth structure. The spurious phase is definitely linked to the microseism excitations in a constrained region of noise sources. In contrast, noise-derived surface waves are related to sources in a broad stationary-phase region. The stationary-phase regions for the noise-derived P waves are not unique (P-PP, PP-PPP, …). It is

potential to monitor the ocean wave activities and microseism excitations in the effective source region with the P-PKPab correlation. The difference in the strength of the spurious phase in the causal and acausal parts of the correlation functions is coincident with the difference in the strength of the causal and acausal microseism sources. That testifies the potentiality. The P-PKPab correlation is not the unique spurious phase emerging in global noise correlations. Multiple spurious arrivals can be observed from the global sections of the noise correlations constructed with both real and synthetic seismograms

(Boué et al., 2013b, 2014; Ruigrok et al., 2008). The double-array slowness analysis and slowness-track method proposed in this paper are also applicable to other noise-derived seismic signals.

*Author contribution.* LL performed the data processing and designed the synthetic experiments. The code for noise data processing was developed based on an early version by PB. All authors contributed to the analysis of the results. LL

prepared the manuscript with contributions from all co-authors. MC provided the funding support leading to this publication.

*Competing interests.* The authors declare that they have no conflict of interest.

*Acknowledgments.* The continuous seismograms of FNET and LAPNET were provided by the National Research Institute

for Earth Science and Disaster Resilience (http://www.fnet.bosai.go.jp) and the Réseau Sismologique & Géodésique Français (http://www.resif.fr), respectively. The global section of earthquake seismograms were obtained from the IRIS GlobalStacks Data Service (https://ds.iris.edu/ds/products/globalstacks/). The global section of synthetic seismograms were obtained from the IRIS Syngine Data Service (https://ds.iris.edu/ds/products/syngine/). The data of synthetic global microseism noise sources were provided by the IOWAGA products (Rascle and Ardhuin, 2013). The Taup program

(Crotwell et al., 1999) and the IASP91 Earth model (Kennett and Engdahl, 1991) were used to calculate the theoretical travel times and ray parameters of seismic phases. The computations were performed with the CIMENT cluster (https://ciment.ujf-grenoble.fr), which is supported by the Rhône-Alpes grant CPER07_13 CIRA (http://www.ci-ra.org). This work is supported by grants from the Simone et Cino Del Duca Foundation, Institut de France, and Labex OSUG@2020 (Investissements d'avenir-ANR10LABX56). The authors also acknowledge Prof. Sidao Ni for comments and suggestions on an early draft

that improve the manuscript.

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





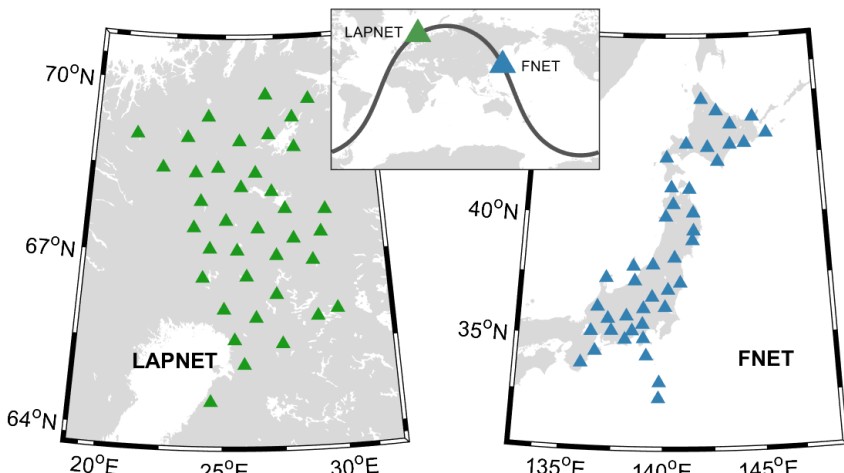

Figure 1: Geographic locations of the 38 stations of the LAPNET array in Finland and the 41 stations of the FNET array in Japan.





**Figure 2: Examples of segment-based noise data processing. A segment with stationary noise and a segment containing a M7.2 teleseism from a daily trace recorded by FNET station BO.YMZ are used for demonstration.**





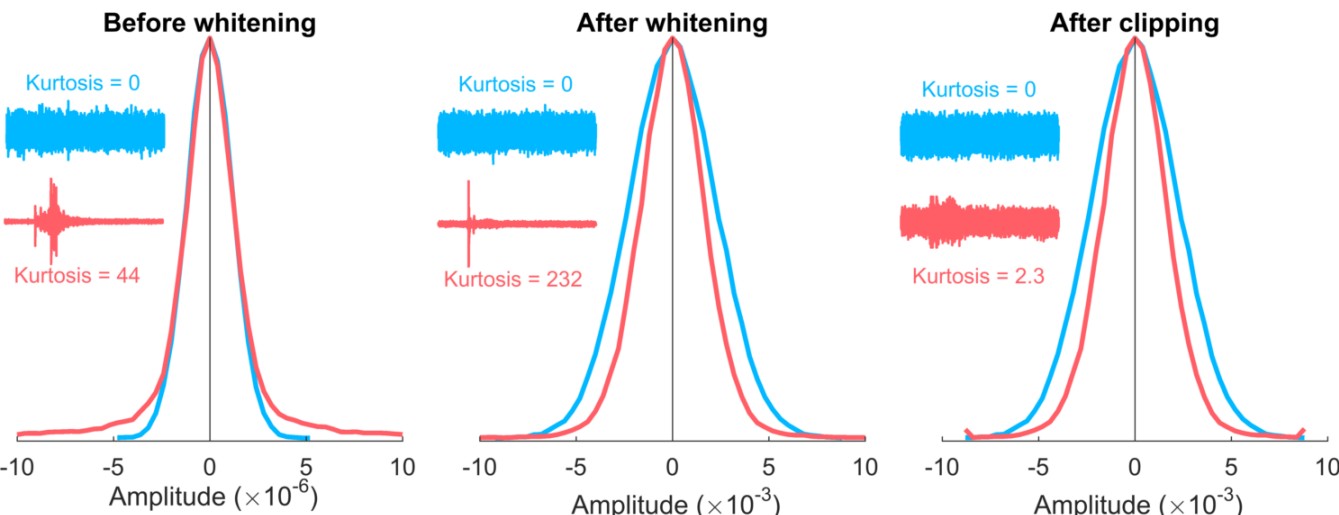

**Figure 3: Kurtosis-based selection filter to determine if a segment contains large-amplitude transients. The two segments used here are the same as in Fig. 2. For display, waveforms are plotted in varying scales. The amplitude histograms are normalized by their own maximums. Histogram tails outside the horizontal axis limits are cropped.**





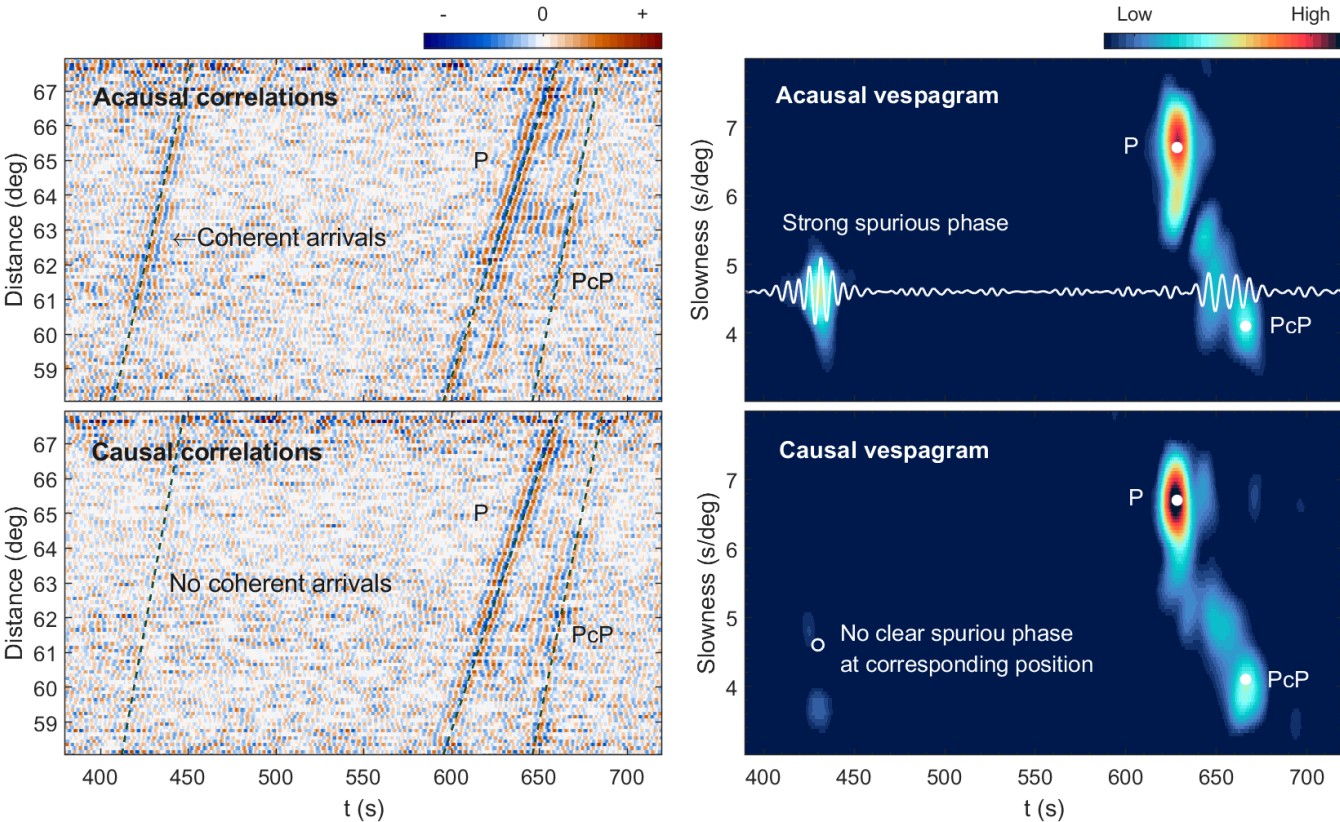

**Figure 4: Waveform sections and vespagrams of the acausal and causal parts of the vertical-vertical noise correlations filtered in the period band from 5 s to 10 s. The acausal section for negative time lags is flipped to share a common time axis with the causal section.**





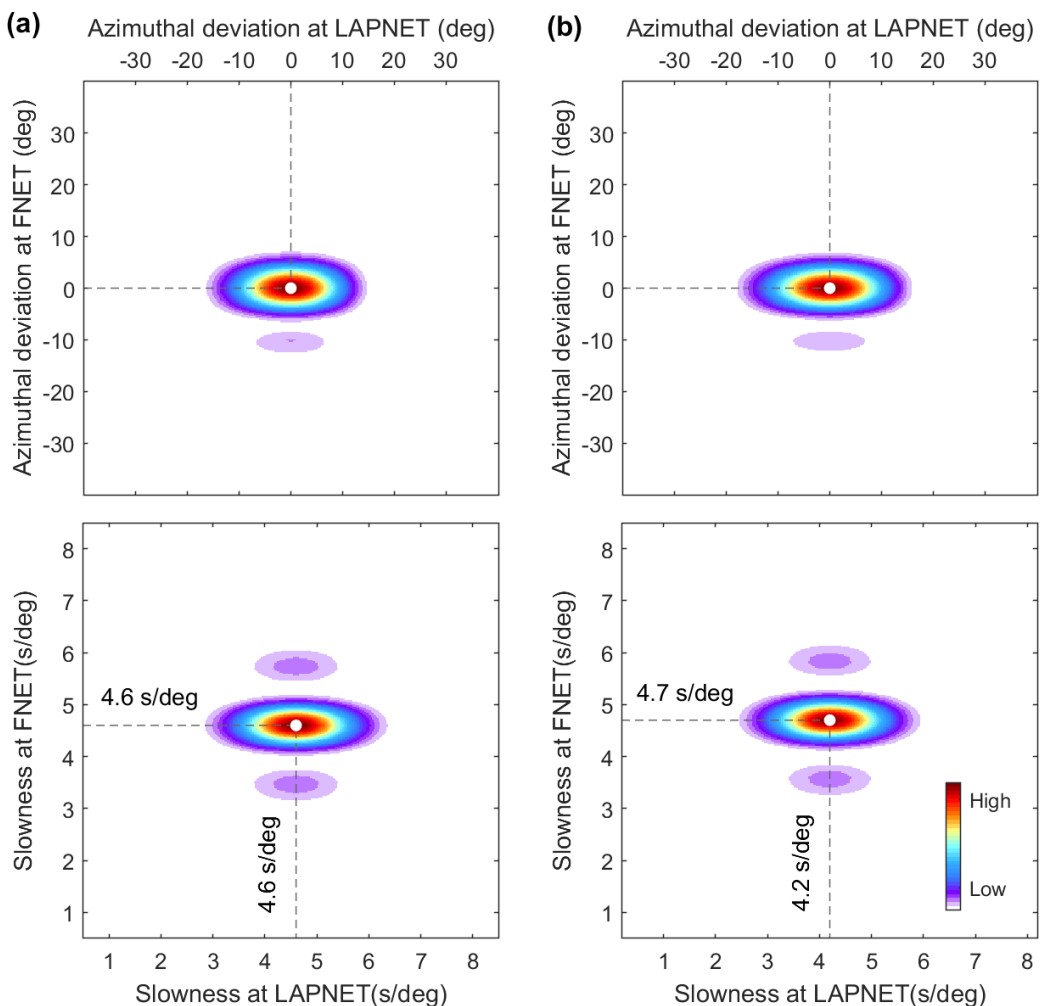

**Figure 5: Numerical tests for the double-array slowness analysis of the FNET-LAPNET correlations at a seismic period of 6.2 s. The input azimuths of the interfering waves are confined to the great circle crossing FNET and LAPNET. The azimuthal deviation refers to the clockwise azimuthal deviation of slowness from the great circle. The input slownesses of the interfering waves are (a) 4.6 s/deg at both LAPNET and FNET and (b) 4.2 s/deg at LAPNET and 4.7 s/deg at FNET, which are well resolved by the slowness analysis.**





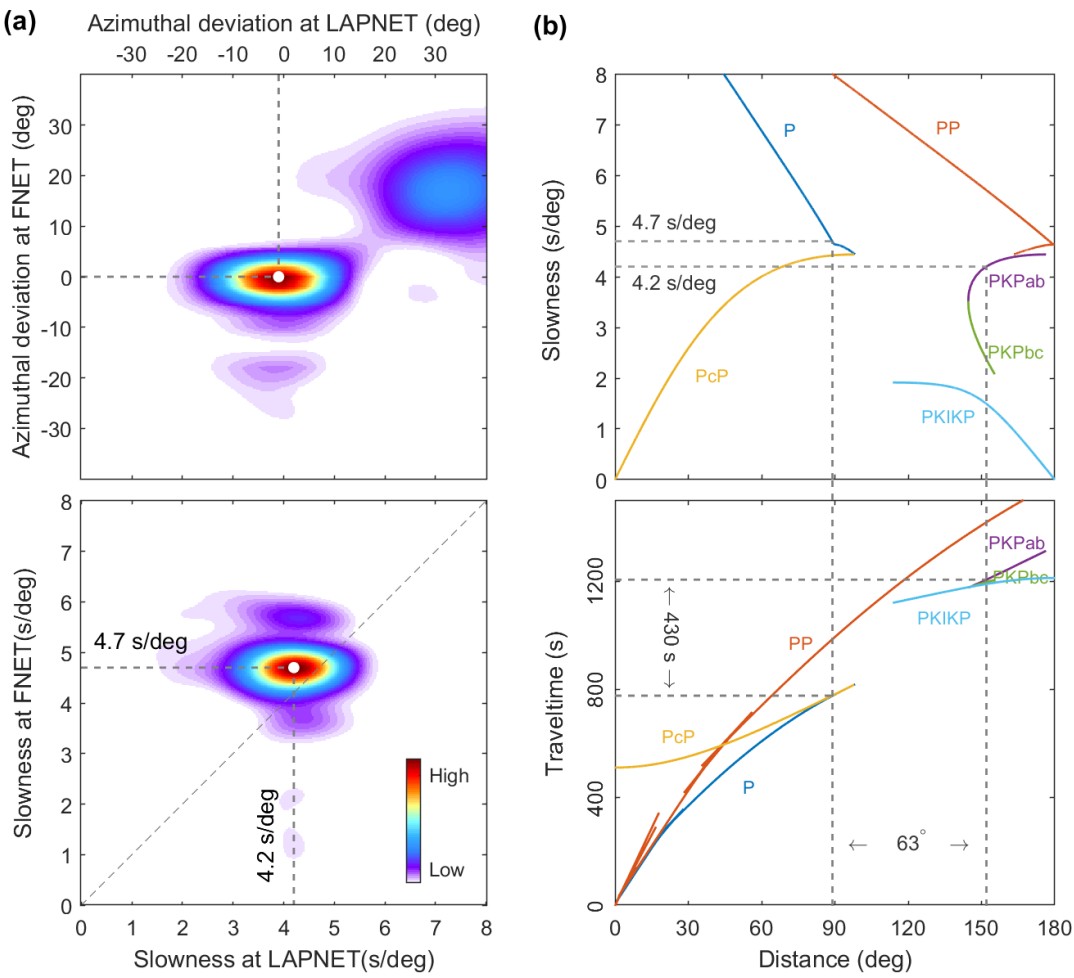

**Figure 6: (a) Results of the double-array slowness analysis for the observed spurious phase. (b) Tracking the interfering waves responsible for the spurious phase using the slowness estimates.**





**Figure 7**: **Global stacks of the vertical components of the seismograms selected from over 2,500 shallow earthquakes (event depth ≤ 50 km and magnitude ≥ 5.4) occurring between 1995 and 2013. The seismograms are filtered around 6 s period and converted into traces of STA/LTA ratios. The STA/LTA traces are binned by epicentral distances in an interval of 0.5° and normalized for plotting. More details can be found on the IRIS Data Services Products website.**





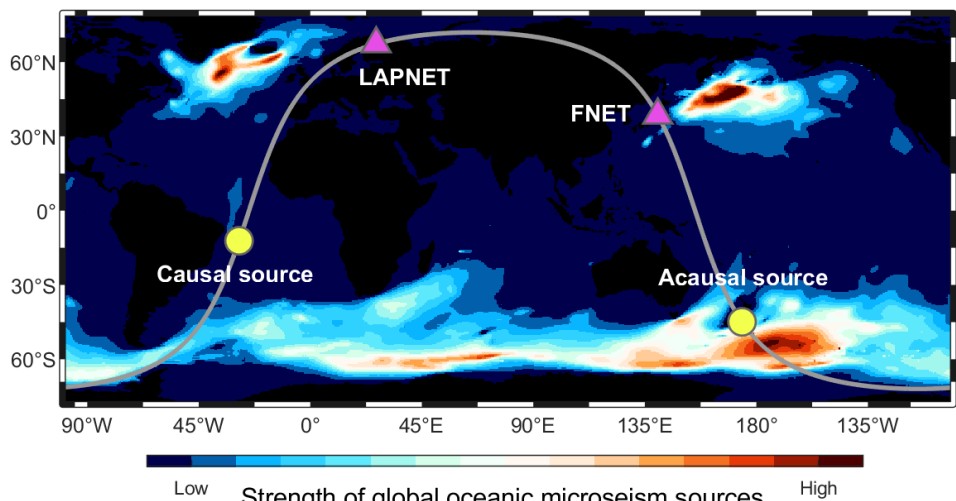

**Figure 8: The global map of oceanic microseism excitation in 2008, at a seismic period of 6.2 s. The source responsible for the acausal spurious phase is located in the ocean south of New Zealand.**

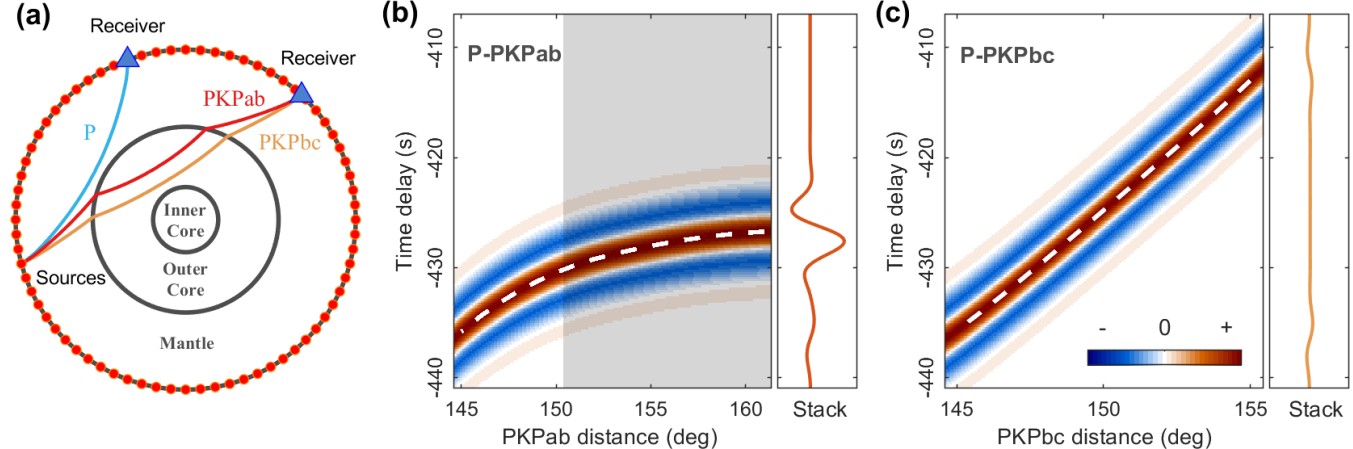

**Figure 9: (a) Geometry of the model to synthesize the inter-receiver correlation function of teleseismic P and PKP waves via source averaging. (b) Source-averaging experiment for the P-PKPab correlations at an inter-receiver distance of 63°. The source-wise P-PKPab correlation functions are synthesized by convolving a 6.2 s period wavelet with the P-PKPab time delays. The final inter-receiver correlation function is obtained by stacking the source-wise correlations. (c) Source-averaging experiment for the P-**
10 **PKPbc correlations.**





**Figure 10: (a)** Global section of synthetic broadband (2 s to 100 s) seismograms obtained from the IRIS Syngine Data Service. **(b)** Seismograms containing P and PKPab waves only, by muting other seismic phases in (a). **(c)** Global section of inter-receiver correlations using the waveform data in (b). **(d)** Global section of synthetic P-PKPab correlations using the ray-based method in Fig. 9.