# Peer review of "Observation and explanation of spurious seismic signals emerging in teleseismic noise correlations"

_Solid Earth, 2019_

## Referee Comment (RC1) · Anonymous Referee #1 · 19 Aug 2019

General comments:

This study investigates the source of spurious arrivals in ambient noise cross-correlation functions calculated over teleseismic differences. The authors explain that such spurious seismic arrivals can be the result of the interference between seismic phases that have time delays that are 'quasi-stationary', that is, their arrival time difference does not vary strongly with source distance. This effect can occur even if the phases do not share a ray path. The authors use two seismic arrays to demonstrate an example involving the P and PKPab phases. In general, this discussion paper is a very nice contribution that will be of interest to a wide audience. I have a few comments that I believe should be addressed before publication, but these are probably quite minor. I will go through these comments in the order in which they appear in the manuscript.

[Figure]

Specific comments:

- In my opinion, the introduction section of this manuscript is a bit thin on relevant detail. Currently, the authors focus on describing the construction of empirical Green's functions, and briefly mention some of the applications. They consign the majority of the detail to a citation for a review paper. I think this approach is fine when it comes to the empirical Green's function approach, as it isn't really the point of this paper, but I do think the introduction should be expanded to provide more background on the spurious arrivals instead.

More specifically, the line of thinking to explain spurious arrivals followed in this paper has already been introduced by Pham et al. (2018), and yet this study has not been cited throughout the current paper. In my opinion, the work of Pham et al. should be presented in the introduction, as it would allow for a nice progression in scientific thinking: Pham et al. focuses on spurious arrivals that share a common ray path, whereas the current study explains those that do not share a ray path.

Pham, T. -S., Tkalčić, H., Sambridge, M., & Kennett, B. L. N. ( 2018). Earth's correlation wavefield: Late coda correlation. Geophysical Research Letters, 45, 3035– 3042. https://doi.org/10.1002/2018GL077244

- Similarly, there should probably be some discussion involving Pham et al. (2018) in section 7.

- This might just be a language issue, but on page 2, line 5 the authors state that there have only been a few noise-derived body wave signals. Whilst body waves are certainly more rare than surface waves, nowadays I don't think you can say there are only a few examples. Some examples that could be cited, including the retrieval of core phases, include but not limited to:

https://doi.org/10.1002/grl.50237

https://doi.org/10.1002/2017GL073230

https://doi.org/10.1093/gji/ggw015

https://doi.org/10.1002/2014GL062198 (Uses the same seismic arrays as the authors)

- On page 3, the authors describe an interesting kurtosis-based method for discarding noise segments that are contaminated by earthquake signals. Is this the first case of this method being used for processing ambient noise? If so, a little bit more clarity is needed. In particular, the 'expectation operator' needs explaining to avoid confusion. Is it some kind of mean? I think if the equation defining kurtosis is properly explained around page 3 line 5, that would be sufficient detail for this paper.

- On page 4 line 20, the authors mention 'numerical experiments'. More detail probably needs to be added here. How were these experiments performed? I assume by simulating plane waves passing over the known array geometries, but it is impossible to tell from the current text.

- Similarly, on page 4 line 20 - 21 the authors quote the simulated slowness values for their numerical experiment, but on the first read it appears as if these values drop out of thin air until it is explained that they match the observed slowness from the real data on line 25. I would suggest that the order of the explanation is changed here so that it is clear that the numerical experiment simulates the observed slowness values.

- On page 5 lines 5 - 15, the authors explain how they identify the relevant interfering phases. In the current form, the explanation is slightly convoluted and hard to follow. I think it would benefit if the authors explicitly state that the culprits are a P-wave sourced 89 degrees from, and recorded at, FNET, and PKPab sourced 152 degrees from LAP-NET. At the moment the arrays at which each phase is recorded is only implied by the text, when it is key to identifying the source region.

- The authors comment on page 5 line 12 that PcP-PKPab also matches the required time delay. Is this candidate discarded due to an incorrect slowness for PcP? Again it isn't stated, but only implied. Perhaps the PcP slowness should be quoted here too to

drive the point home.

- A minor confusion occurs on page 5 line 17, the authors state that Fig. 6 can be used to located the source responsible, when in reality Fig. 6 only gives you the source distances. Unless I'm mistaken, to actually locate the source you need other information such as the array locations, and whether the source is causal or acausal.

- The supplementary material is currently just a pile of a couple of figures referred to in the main text. I think the supplementary material should include the information required to stand on its own. I think a couple of sentences explaining each supplemental figure, and its relevance to the main text, are warranted.

Technical comments: - On page 8 lines 4 - 7 there are a few sentences that don't make much sense, and are grammatically incorrect. I suggest the authors rewrite these sentences to clarify.

- In Fig. 4, on the bottom vespagram, 'spurious' is missing an 's'.

In conclusion, I believe that in order to provide the clarifications and explanations that I have requested above, it is likely only a minor revision will be required.
* * *

---

## Referee Comment (RC2) · Pilar Sánchez Sánchez-Pastor (Referee) · 29 Oct 2019

This is an interesting manuscript that studies a spurious signal observed in the correlograms of seismic noise between two distant seismic networks. The authors employ the double-beam method to estimate the slowness of several seismic phases as a function of distance and thus, track the observed interfering waves and determine the origin of that spurious signal. Furthermore, the authors provide a physical explanation for such signal through numerical simulations and observe it as well in synthetic correlograms.

In my opinion, the study is well addressed and scientifically valuable. However, the presentation of the manuscript should be improved before possible publication. Basically, the manuscript needs to be written with more care and some minor corrections

are required. My suggestions and comments are described below.

- In the Introduction, I would have liked a better introduction of spurious signals, why is useful to study them, mention the previous similar studies and, in general, explain better the problematic. Also, I would update some references with new studies and add some in pag. 2, lines 4-9.

- Line 29, pag. 2: Vague sentence. Some readers very likely would not understand what you mean.

- Line 30, pag. 2: It is worth it to specify the amplitude threshold (how many times of the standard deviation) that you consider to clip the waveforms and avoid large transients.

- Line 5, pag. 3: If it is the first time that the kurtosis is employed in seismic noise processing, the authors should explain it better. For example, the equation described is a comparison between the kurtosis of the distribution under study and the kurtosis of a normal distribution, which is 3.

- Line 8, pag. 3: "the segments beyond 1.5 are discarded" why this value? It would be proper a short comment to explain it.

- Line 20, pag. 4: Vague sentence. Which numerical experiments?

- Lines 4-9, pag. 5: I think the proposed slowness-track method to identify the ray paths of the interfering waves is not enough clear. In my opinion, this paragraph can be improved and make easier to follow the idea.

- Line 7, pag. 5: "The pairs of seismic phases are rejected if the difference between the distances from the source to the receivers differs from 63°or if their time delay deviates from 430 s" why? It could be obvious but indicating a reason works out well for a better understanding.

- Line 20, pag. 7: I imagine those results imply a lot of work and they are interesting. So perhaps it is worth adding a supplementary figure.

FIGURES:

- Figure 3: you should use same colours as in Fig 2 to be consistent. Also, the title "after clipping" I would say amplitude clipping or something similar in order to avoid misunderstandings.

- Figure 4: The labels a) b) etc are missing. Moreover, you should explain the overlapped signal in the figure caption.

- Figure 5: From my point of view, it can be added to the supplementary material. If you consider the supplement is already too long, the Figure S1 is dispensable.

- Figure S1: I would change "removed-mean series" for pre-processed series because you correlate after removing the mean, trend, filtering, whitening... Moreover, I would add in the colored bars at the top a label "i" and in the bars at the bottom "i" and "i-$\tau$" (following the notation of the eq) to illustrate the dislocation applied by the correlation. Although, I believe it is better only correlate the "effective samples" instead of adding zeros. In this way, for large lag times you underestimate the correlation.

- Figure 8: is it computed or taken from other study?

- Figure 10: you should describe what the red point represents in the figure caption even if this seems obvious. In my view, figure captions should be auto-explicative and if they are not, one should indicate where the reader could find the information.

OTHER MINOR COMMENTS:

- Line 2, pag. 2: "We refer to (Campillo and Roux, 2015)" without parenthesis.

- Lines 26-29, pag. 5: reference?

- Line 17-20, pag. 6: Those sentences can be improved.

- Line 29, pag. 6: "The ray-based simulation above" would be better like: The above-described ray-based simulation...

- Line 23, pag. 7: In my opinion, this section should be called 'Conclusions'.

---

## Author Comment (AC1) · 26 Nov 2019

General comments:

This study investigates the source of spurious arrivals in ambient noise cross-correlation functions calculated over teleseismic differences. The authors explain that such spurious seismic arrivals can be the result of the interference between seismic phases that have time delays that are 'quasi-stationary', that is, their arrival time difference does not vary strongly with source distance. This effect can occur even if the phases do not share a ray path. The authors use two seismic arrays to demonstrate an example involving the P and PKPab phases. In general, this discussion paper is a very

nice contribution that will be of interest to a wide audience. I have a few comments that I believe should be addressed before publication, but these are probably quite minor. I will go through these comments in the order in which they appear in the manuscript.

==========

Reply: The authors would like to thank the referee for the careful review and helpful suggestions on the manuscript. We modified the manuscript accordingly. Point-to-point responses are provided below.

==========

Specific comments:

- In my opinion, the introduction section of this manuscript is a bit thin on relevant detail. Currently, the authors focus on describing the construction of empirical Green's functions, and briefly mention some of the applications. They consign the majority of the detail to a citation for a review paper. I think this approach is fine when it comes to the empirical Green's function approach, as it isn't really the point of this paper, but I do think the introduction should be expanded to provide more background on the spurious arrivals instead. More specifically, the line of thinking to explain spurious arrivals followed in this paper has already been introduced by Pham et al. (2018), and yet this study has not been cited throughout the current paper. In my opinion, the work of Pham et al. should be presented in the introduction, as it would allow for a nice progression in scientific thinking: Pham et al. focuses on spurious arrivals that share a common ray path, whereas the current study explains those that do not share a ray path. Pham, T. -S., TkalËĞ ci′ c, H., Sambridge, M., & Kennett, B. L. N. (2018). Earth's correlation wavefield: Late coda correlation. Geophysical Research Letters, 45, 3035– 3042. https://doi.org/10.1002/2018GL077244 - Similarly, there should probably be some discussion involving Pham et al. (2018) in section 7.

==========

Reply: The authors recognize that the initial version of Introduction needs to be extended. We thank the review for this comment. We were aware of the work by Pham et al. (2018) that interpreted spurious phases in earthquake coda correlations with the stationary-phase arguments: "all phases identified in the correlation wavefield correspond to differences between seismic arrivals with the same ray parameter and a subset of propagation legs in common". We initially thought that readers would be confused by an introduction on coda wave interferometry, while we only focus on microseism noise correlations. Ambient wavefields are dominated by ballistic waves from oceanic microseism sources (5 to 10 s periods). Coda waves excited by large earthquakes are dominated by high-order modes at longer periods (> 20 s) and corresponding to core-related reverberations. We notice that it has not been pointed out explicitly in existing literatures that at large scale, ambient noise correlations are distinct from earthquake coda correlations. Not mentioning the latter could also be misleading. As suggested by the reviewer, we have modified the Introduction and Conclusion sections. We have also added a new Fig. S5 that demonstrates the difference between microseism correlations and coda correlations.

==========

- This might just be a language issue, but on page 2, line 5 the authors state that there have only been a few noise-derived body wave signals. Whilst body waves are certainly more rare than surface waves, nowadays I don't think you can say there are only a few examples. Some examples that could be cited, including the retrieval of core phases, include but not limited to: https://doi.org/10.1002/grl.50237 https://doi.org/10.1002/2017GL073230 https://doi.org/10.1093/gji/ggw015 https://doi.org/10.1002/2014GL062198 (Uses the same seismic arrays as the authors)

==========

Reply: The reviewer is correct. It was a typo. We meant fewer compared to surface

waves. Several new citations have been added to the Introduction.

==========

- On page 3, the authors describe an interesting kurtosis-based method for discarding noise segments that are contaminated by earthquake signals. Is this the first case of this method being used for processing ambient noise? If so, a little bit more clarity is needed. In particular, the 'expectation operator' needs explaining to avoid confusion. Is it some kind of mean? I think if the equation defining kurtosis is properly explained around page 3 line 5, that would be sufficient detail for this paper.

==========

Reply: To our knowledge, it is the first time that the kurtosis has been applied to noise data processing. The reviewer is right that the expectation here refers to the mean value. We clarify it being "arithmetic mean" in the revision.

==========

- On page 4 line 20, the authors mention 'numerical experiments'. More detail probably needs to be added here. How were these experiments performed? I assume by simulating plane waves passing over the known array geometries, but it is impossible to tell from the current text.

==========

Reply: The referee assumed correctly. We have added some details on the numerical experiments. "To investigate the resolution capability of the double-array slowness analysis for the FNET-LAPNET geometry, we make numerical experiments by presuming (a) the same slowness at FNET and LAPNET (4.6 s/deg), and (b) different slownesses at FNET (4.7 s/deg) and LAPNET (4.2 s/deg). Assuming plane waves passing through FNET and LAPNET, the time delays between FNET and LAPNET station pairs can be calculated from Eq. (1). The wavelet of the observed spurious phase (5 to 10 s bandpass filtered) is convolved with the time delays to synthesize the correlation

functions. The synthesized correlations are beamed by Eq. (2) for various slownesses. The results are plotted in Fig. 6. In both cases, the slownesses of the correlated waves at FNET and LAPNET are well resolved, justifying the reliability of our slowness discrepancy estimation in Fig. 5a."

==========

- Similarly, on page 4 line 20 - 21 the authors quote the simulated slowness values for their numerical experiment, but on the first read it appears as if these values drop out of thin air until it is explained that they match the observed slowness from the real data on line 25. I would suggest that the order of the explanation is changed here so that it is clear that the numerical experiment simulates the observed slowness values.

==========

Reply: We agree that exchanging figs 5 and 6 and relevant text improves readability. Thanks for this suggestion.

==========

- On page 5 lines 5 - 15, the authors explain how they identify the relevant interfering phases. In the current form, the explanation is slightly convoluted and hard to follow. I think it would benefit if the authors explicitly state that the culprits are a P-wave sourced 89 degrees from, and recorded at, FNET, and PKPab sourced 152 degrees from LAP-NET. At the moment the arrays at which each phase is recorded is only implied by the text, when it is key to identifying the source region.

==========

Reply: We agree with the proposed clarification and have modified the statement accordingly.

==========

- The authors comment on page 5 line 12 that PcP-PKPab also matches the required

time delay. Is this candidate discarded due to an incorrect slowness for PcP? Again it isn't stated, but only implied. Perhaps the PcP slowness should be quoted here too to drive the point home.

==========

Reply: Yes, the slowness comparisons are critical. We have clarified this point and quoted the PcP slowness.

==========

- A minor confusion occurs on page 5 line 17, the authors state that Fig. 6 can be used to located the source responsible, when in reality Fig. 6 only gives you the source distances. Unless I'm mistaken, to actually locate the source you need other information such as the array locations, and whether the source is causal or acausal.

==========

Reply: Fig. 6 provides source-receiver distances. Receiver locations are of course necessary for locating the sources. We have clarified that in the revision.

==========

- The supplementary material is currently just a pile of a couple of figures referred to in the main text. I think the supplementary material should include the information required to stand on its own. I think a couple of sentences explaining each supplemental figure, and its relevance to the main text, are warranted.

==========

Reply: Subheadings and explanatory sentences have been added to Supplementary.

==========

Technical comments: - On page 8 lines 4 - 7 there are a few sentences that don't make much sense, and are grammatically incorrect. I suggest the authors rewrite these

sentences to clarify.

==========

Reply: We have rewritten this part. Thanks for pointing this out.

==========

- In Fig. 4, on the bottom vespagram, 'spurious' is missing an 's'.

==========

Reply: Corrected. Thanks.

==========

In conclusion, I believe that in order to provide the clarifications and explanations that I have requested above, it is likely only a minor revision will be required.

---

## Author Comment (AC2) · 26 Nov 2019

Pilar Sánchez Sánchez-Pastor (Referee #2)

This is an interesting manuscript that studies a spurious signal observed in the correlograms of seismic noise between two distant seismic networks. The authors employ the double-beam method to estimate the slowness of several seismic phases as a function of distance and thus, track the observed interfering waves and determine the origin of that spurious signal. Furthermore, the authors provide a physical explanation for such signal through numerical simulations and observe it as well in synthetic correlograms. In my opinion, the study is well addressed and scientifically valuable. However, the presentation of the manuscript should be improved before possible publication. Basi-

cally, the manuscript needs to be written with more care and some minor corrections are required. My suggestions and comments are described below.

==========

Reply: The authors would like to thank the reviewer for her careful reading and constructive comments. Point-to-point replies are provided below.

==========

- In the Introduction, I would have liked a better introduction of spurious signals, why is useful to study them, mention the previous similar studies and, in general, explain better the problematic. Also, I would update some references with new studies and add some in pag. 2, lines 4-9.

==========

Reply: The authors recognize that the initial version of Introduction needs to be extended. We thank the reviewer for this comment. The other reviewer made a similar comment on the Introduction. We have added new citations and extended the Introduction to better describe the background, especially, some existing applications of noise-derived deep body waves (including spurious phase).

==========

- Line 29, pag. 2: Vague sentence. Some readers very likely would not understand what you mean.

==========

Reply: We agree with the reviewer and have removed this dispensable sentence.

==========

- Line 30, pag. 2: It is worth it to specify the amplitude threshold (how many times of the standard deviation) that you consider to clip the waveforms and avoid large transients.

==========

Reply: We mention in the revision that we clipped at 3.8*std, which is just an empirical choice following previous studies (Poli et al., 2012; Boué et al., 2013). We did not specify the value because the choice is more or less arbitrary. No problem to choose other values. Of course, a very large value (e.g., 100 times) would make the clipping ineffective in removing impulses. A very small value (e.g., 0.1 times) would have a similar effect as the one-bit resampling. A modest choice of 3.8 leads to an effective clipping of large transients and retains the waveform of stationary noise (Fig. 3 for examples).

==========

- Line 5, pag. 3: If it is the first time that the kurtosis is employed in seismic noise processing, the authors should explain it better. For example, the equation described is a comparison between the kurtosis of the distribution under study and the kurtosis of a normal distribution, which is 3.

==========

Reply: Following comments by both reviewers, we have described more on kurtosis, and also explained, as suggested here, that including the term 3 makes the kurtosis of Gaussian distribution zero.

==========

- Line 8, pag. 3: "the segments beyond 1.5 are discarded" why this value? It would be proper a short comment to explain it.

==========

Reply: We have clarified in the revision the threshold of 1.5 is empirical. The threshold, if too small (below ∼1), will reject good noise segments, and if too large (above 3-5), will let pass impulsive segments. A value between 1 and 2 is suggested. From our

experience, 1.5 works fine for various datasets.

==========

- Line 20, pag. 4: Vague sentence. Which numerical experiments?

==========

Reply: This problem was also raised by the other reviewer. We have described more on the numerical experiments in the revision: "To investigate the resolution capability of the double-array slowness analysis for the FNET-LAPNET geometry, we make numerical experiments by presuming (a) the same slowness at FNET and LAPNET (4.6 s/deg), and (b) different slownesses at FNET (4.7 s/deg) and LAPNET (4.2 s/deg). Assuming plane waves passing through FNET and LAPNET, the time delays between FNET and LAPNET station pairs can be calculated from Eq. (1). The wavelet of the observed spurious phase (5 to 10 s bandpass filtered) is convolved with the time delays to synthesize the correlation functions. The synthesized correlations are beamed by Eq. (2) for various slownesses."

==========

- Lines 4-9, pag. 5: I think the proposed slowness-track method to identify the ray paths of the interfering waves is not enough clear. In my opinion, this paragraph can be improved and make easier to follow the idea. - Line 7, pag. 5: "The pairs of seismic phases are rejected if the difference between the distances from the source to the receivers differs from 63 âŮę or if their time delay deviates from 430 s" why? It could be obvious but indicating a reason works out well for a better understanding.

==========

Reply: We agree with these comments and have rephrased this part.

==========

- Line 20, pag. 7: I imagine those results imply a lot of work and they are interesting.

So perhaps it is worth adding a supplementary figure.

==========

Reply: See the new Fig. S6 and relevant supplemental text. Thanks for this suggestion.

==========

FIGURES: - Figure 3: you should use same colours as in Fig 2 to be consistent. Also, the title "after clipping" I would say amplitude clipping or something similar in order to avoid misunderstandings.

==========

Reply: Modified accordingly. Thank you for pointing this out. We ignored this detail.

==========

- Figure 4: The labels a) b) etc are missing. Moreover, you should explain the over-lapped signal in the figure caption.

==========

Reply: Labels and text annotation for the beamed signal have been added.

==========

- Figure 5: From my point of view, it can be added to the supplementary material. If you consider the supplement is already too long, the Figure S1 is dispensable.

==========

Reply: Thanks for your suggestion. Considering that double-array slowness analysis is first proposed in this paper and the resolution is critical to our argument that the interfering waves have distinct slownesses, and also that SE is an electronic journal that has no constrains on the paper length, we prefer to keep the figure in the main text.

==========

- Figure S1: I would change "removed-mean series" for pre-processed series because you correlate after removing the mean, trend, filtering, whitening... Moreover, I would add in the colored bars at the top a label "i" and in the bars at the bottom "i" and "i-$\tau$" (following the notation of the eq) to illustrate the dislocation applied by the correlation. Although, I believe it is better only correlate the "effective samples" instead of adding zeros. In this way, for large lag times you underestimate the correlation.

==========

Reply: Figure S1 is intended to explain the computation of correlation function in a general sense. The formulae displayed in the figure only require the series being demeaned. So, it is not a problem. Of course, the referee is right in the context of seismograms. The correlation function is routinely calculated by FFT for efficiency. The zero padding at the bottom of Fig. S1 is just for explanatory purpose. Concerning the underestimation of correlation function at large lags, a modified scheme for calculating the correlation function described in figure 2.21 in section 2.4.2 of my thesis can deal with this problem (https://www.theses.fr/2018GREAU023.pdf). This is irrelevant to the topic of this paper. So, we do not talk much on it.

==========

- Figure 8: is it computed or taken from other study?

==========

Reply: All data sources are indicated in the Acknowledgement. We have clarified in the caption that the data in Fig. 8 come from Rascle and Ardhuin (2013).

==========

- Figure 10: you should describe what the red point represents in the figure caption even if this seems obvious. In my view, figure captions should be auto-explicative and

if they are not, one should indicate where the reader could find the information.

==========

Reply: We agree that it is common to describe lines and symbols in the caption. But this appears discouraged by Solid Earth. "A legend should clarify all symbols used and should appear in the figure itself, rather than verbal explanations in the captions (e.g. "dashed line" or "open green circles")." (https://www.solid-earth.net/for_authors/manuscript_preparation.html) We used text annotations on the figure to indicate that the red dot corresponds to the observed spurious phase beamed at 63° distance.

==========

OTHER MINOR COMMENTS: - Line 2, pag. 2: "We refer to (Campillo and Roux, 2015)" without parenthesis.

==========

Reply: Corrected. Thanks.

==========

- Lines 26-29, pag. 5: reference?

==========

Reply: Added.

==========

- Line 17-20, pag. 6: Those sentences can be improved.

==========

Reply: We have rephrased this part.

==========
- Line 29, pag. 6: "The ray-based simulation above" would be better like: The above-described ray-based simulation... - Line 23, pag. 7: In my opinion, this section should be called 'Conclusions'.

==========

Reply: Modified accordingly. Thanks.

==========